# Potential of a Small Molecule Carvacrol in Management of Vegetable Diseases

**DOI:** 10.3390/molecules24101932

**Published:** 2019-05-20

**Authors:** Qingchun Liu, Kang Qiao, Shouan Zhang

**Affiliations:** 1Tropical Research and Education Center, Department of Plant Pathology, University of Florida, IFAS, Homestead, FL 33031, USA; q.liu1@ufl.edu (Q.L.); qiaokang@ufl.edu (K.Q.); 2Key Laboratory of Pesticide Toxicology & Application Technique, College of Plant Protection, Shandong Agricultural University, Tai’an 271018, Shandong, China

**Keywords:** carvacrol, small molecule, vegetable pathogens, IPM

## Abstract

Carvacrol, a plant-derived volatile small molecule, is effective against various agents that can cause damage to humans, the food processing industry, and plants, and is considered a safe substance for human consumption. In this short communication, previous studies on the effectiveness of carvacrol against various agents, particularly plant pathogens and their associated mechanisms are described. In our study, carvacrol was found to be effective on media against several soilborne pathogens and in planta against three foliar pathogens (*Xanthomonas perforans*, *Alternaria tomatophila*, and *Podosphaeraxanthii*) of important vegetable crops in south Florida of the United States. Current research findings indicated that the effectiveness of carvacrol against various plant pathogens tested was associated with its direct bactericidal/fungicidal effect, which was affected greatly by its volatility. Development of new formulations to overcome the volatility and to prolong the effectiveness of carvacrol was also presented. Our studies on carvacrol suggested that, with advanced development of new formulations, carvacrol could be used as a promising tool in the integrated pest management for bacterial, fungal, and viral pathogens of important vegetable crops in Florida, the USA, and the world.

## 1. Carvacrol

Carvacrol, 2-methyl-5-(1-methylethyl)-phenol, is a small molecule present in many plant species of the Lamiaceae family, including oregano (*Origanum vulgare*) and thyme (*Thymus vulgaris*). Carvacrol is the major component in essential oils of these plants, particularly those from oregano [1,2], and is a volatile secondary metabolite that is liquid at room temperature, insoluble in water but soluble in ethanol [2,3,4]. Carvacrol has one single hydroxyl group (–OH), which is next to the methyl group in the aromatic ring. The unique position of the –OH group in carvacrol plays a critical role in its chemical and biological characteristics [1,5]. Carvacrol is considered as a safe substance being used for human consumption in the United States and globally [1].

## 2. Antimicrobial Activity of Carvacrol and Associated Mechanisms

Carvacrol has been studied extensively for its antimicrobial activity in the medical field and the food processing industry, where the majority of the targets are bacteria and fungi [1,2,3,6,7]. The antimicrobial activity of carvacrol, techniques that can be used in assessing its efficacy, its modes of action, interaction with other agents, and formulation development for its effective application have been reviewed in many publications [1,2,3,6]. In addition to bacterial and fungal microorganisms, carvacrol was also found to be effective against viral particles [8,9] as well as larvae and adult insects [10,11,12]. 

Many studies conducted to address the modes of action of carvacrol against various bacteria and fungi have already been reviewed [1,2,3]. It was found that the hydrophobic nature of the –OH group in carvacrol as well as its ability to exchange its protons play critical roles in its antimicrobial activities. These factors have been shown to influence the cell wall and membrane integrity of microbial cells. In addition, carvacrol can affect physiological processes inside the cell, including binding to DNA and blocking ergosterol biosynthesis. When it is applied against bacteria, carvacrol can influence virulence factors, including reduced toxin production and biofilm formation by the bacteria. Most recent studies indicated that carvacrol affects gene expression [13,14] and subsequently reduces virulence of microorganisms [15,16,17].

## 3. Studies on Plant Pathogens and Research Findings Against Vegetable Diseases Involving Carvacrol 

In plant science, there are some studies on the effect of carvacrol against plant pathogens, including soil-borne pathogens, plant-parasitic nematodes, foliar pathogens, and postharvest pathogens (Table 1). However, fewer studies have been conducted to evaluate its effect on plant diseases of the above-ground parts of the plants [18,19]. Recently, small molecules including carvacrol have been studied by our group for their potential in managing important diseases on tomato and squash. The goal of this short communication was to present the critical findings of our studies on carvacrol, to discuss future developments on effective applications of carvacrol in managing plant pathogens, and to explore its potential as an alternative tool in management of main diseases of vegetable crops important in south Florida of the United States. 

In studying carvacrol for its antimicrobial activity, the widely used agar diffusion method is considered inappropriate due to its volatile character, whereas the agar broth or liquid broth cultures are recommended [3]. In our study, it was found that carvacrol was effective in inhibiting mycelial growth of *Phytophthora capsici* on agar media, however, its efficacy decreased by about 50% when 1-week old media amended with carvacrol was used compared to those freshly prepared at the day (Figure 1). Therefore, care needs to be taken for the age of the media after carvacrol is amended into the media when carvacrol is assessed for its inhibitory effect on mycelial growth of fungi or fungal-like microorganisms. 

Most studies on essential oils and their major components including carvacrol in the reviews mentioned previously were conducted *in vitro*. In our study, carvacrol was found to be effective against pathogens isolated from vegetable crops in south Florida. Carvacrol significantly (*P* < 0.05) reduced mycelia growth of *Fusarium* spp. and *Rhizoctonia solani* (from tomato) at 0.25 mM, and completely inhibited mycelial growth at 0.5 mM (Table 2). Against an isolate of *Phytophthora capsici* (from squash), carvacrol reduced mycelial growth by 70% and 91% at 0.25 and 0.5 mM, respectively, and completely inhibited mycelial growth at 1.0 mM (Figure 1). In nutrient broth, carvacrol showed inhibitory effect at 0.5 mM or lower concentrations on the population growth of *Xanthomonas perforans*, the causal pathogen of bacterial spot on tomato in Florida. Bactericidal effect was first shown at 0.5 mM, and clearly detected at 0.75 mM and greater concentrations (Table 2). Both inhibitory and bactericidal effects were dose-related. These results indicated that carvacrol can be effective at very low concentrations against both fungal and bacterial pathogens, and can be used as an alternative to synthesized chemicals in control of major diseases of vegetable crops with development of suitable formulations, particularly for organic vegetable production.

Carvacrol was earlier found to be effective in reducing bacterial populations within 1 h or shorter time of contact [32,33]. Interestingly, in our study, similar results were also found with carvacrol, and the population of *X. perforans* was reduced to undetectable levels after 2 h incubation with carvacrol at 0.75 mM (Table 2). In another study, carvacrol at 0.5% for 1 minute during a flume-tank-washing process reduced populations of *Escherichia coli* O157:H7 to undetectable levels on three leafy vegetables [34]. Such treatments could be applied potentially as an alternative to bleach used in processing water in packing house for fresh market tomato and other tropical fruit to reduce incidence of fruit decay during storage and shipment. Furthermore, such inhibitory or bactericidal /fungicidal effects of carvacrol can be enhanced in an acidic condition or in combination with another chemical such as nisin [35,36].

Application of essential oils, in which carvacrol is the major component, has also been shown to be effective in reducing severity of foliar diseases under field conditions, including bacterial spot (*Xanthomonas* spp.) and bacterial speck (*Pseudomonas syringae* pv. *tomato*) on tomato [19] and powdery mildew (*Podosphaera xanthii*) on zucchini [18]. In our greenhouse study, carvacrol had no effects on squash powdery mildew (*P. xanthii*) when it was applied as a foliar spray at 2 h before the pathogen inoculation. However, an additional treatment with carvacrol at 0.1, 0.5, 1.0, and 5.0 mM 24 h after inoculation significantly reduced the disease compared to the untreated control (Table 3). These results indicated that carvacrol had great potential in reducing plant diseases when it was applied as foliar sprays if its effectiveness can be maintained for 1 h because of its quick action against the pathogens (Table 2 and Table 3). In a field trial conducted in Homestead, Florida in 2018, carvacrol at both concentrations of 0.5 and 1.0 mM significantly reduced disease severity of bacterial spot in tomato compared to the untreated control (Table 4). In addition, carvacrol was also found to decrease severity of early blight (*Alternaria tomatophila*) in the same tomato field trial. Furthermore, carvacrol significantly improved plant vigor and increased fruit yield compared to the untreated control. Results from this field trial were very encouraging, though the field trial with carvacrol will need to be repeated in the future. 

Even though carvacrol showed excellent effects against various plant pathogens, its practical application in the field for control of plant diseases is limited due to its volatility, which was also reflected by the results from our greenhouse study on squash powdery mildew, in which carvacrol was applied before inoculation (Table 3). Much of current effort is to develop new formulations to ensure its effectiveness by prolonging its presence through special formulations, such as slow-release nanoparticle formulations of carvacrol. Our results on squash powdery mildew indicated that presence of carvacrol during the pathogen infection period could significantly reduce disease severity (Table 3), which supports the effectiveness of such strategy. Most recent developments on nanoparticles containing carvacrol, particularly the development with materials of low toxicity and high biocompatibility for safe nanoparticles, have been reviewed [1,6,37,38,39]. One encouraging finding was that the antifungal effects of carvacrol against spore germination and fungal growth of *Aspergillus niger* were extended to 30 days, when it was loaded into silica mesoporous supports [40]. Such advances in extending the effectiveness of carvacrol through new formulations could make it more applicable and effective for carvacrol in managing important diseases of vegetable crops in south Florida. In addition, prolonged presence of carvacrol on tomato plants through application of such new formulations could make it a tool for managing important viral pathogens such as tomato chlorotic spot virus (TCSV) in south Florida [41]. TCSV was first detected from tomato fields in Homestead, Florida in 2012 [41]. It is transmitted by thrips and has become a devastating tospovirus threatening tomato production in south Florida [42]. It was reported that carvacrol can affect the behavior of thrips [11], carvacrol could be promising in integrated management programs for this economically important virus in tomato production in Florida. 

In conclusion, carvacrol was found to be effective against many plant pathogens in previous studies and several plant pathogens of important vegetable crops in south Florida in our study. Further studies are needed to overcome its volatility, including new formulation development, therefore prolong its effectiveness in reducing disease severity. The potential for carvacrol being included as a tool for developing integrated pest management strategy is promising. 

## Figures and Tables

**Figure 1 molecules-24-01932-f001:**
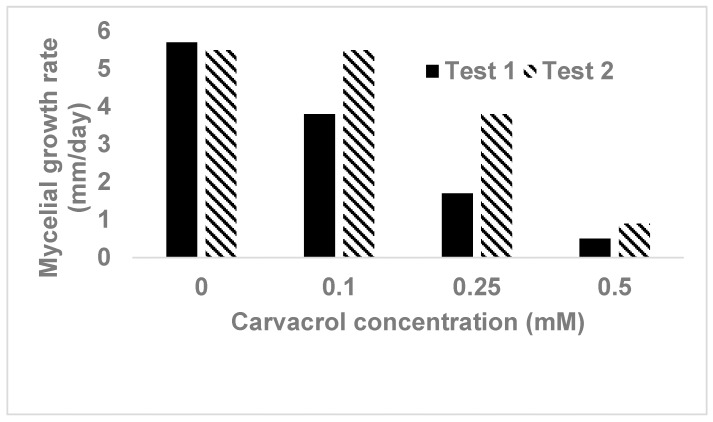
Effect of carvacrol on mycelial growth of *Phytophthora capsici* on V8 juice agar. Carvacrol was added into media right before the media was poured into Petri dishes. Test 1 was conducted with inoculating media on the same day of its preparation, whereas in the test 2 inoculation was carried on 7-day-old agar plates. No mycelial growth was observed on the plates containing carvacrol at 1.0 mM in both tests.

**Table 1 molecules-24-01932-t001:** Summary of studies on carvacrol or essential oils for control of plant diseases.

Pathogen	Crop/Disease	Application Method	Studies Conducted	Reference
*Podosphaera xanthii* fungus	Zucchini/powdery mildew	Foliar spray	*In vivo*	Donnarumma et al., 2015 [18]
*Xanthomonas* spp.; *Pseudomonas syringae* pv. *tomato* bacterium	Tomato/bacterial spot; bacterial speck	Foliar spray	*In vivo*	Giovanale et al., 2017 [19]
*Phytophthora capsici* oomycete	Zucchini/crown rot	Amended in soil	*In vitro* and *in vivo*	Bi et al., 2012 [20]
*Rhizoctonia solani* fungus	Tomato/damping-off	Amended in soil	*In vivo*	Gwinn et al., 2010 [21]
*Sclerotiniasclerotiorum* fungus	Tomato	Amended in soil	*In vitro* and *in vivo*	Soylu et al., 2007 [22]
*Ralstonia solanacearum* bacterium	Tomato/bacterial wilt	Amended in soil	*In vivo*	Pradhanang et al., 2003 [23]
Various plant-parasitic nematodes	English boxwood	Amended in soil	*In vivo*	Pérez and Lewis, 2006) [24]
*Meloidogyne javanica* nematode	Cucumber/root-knot nematode	Amended in soil	*In vitro* and *in vivo*	Oka et al., 2000 [25]
*Meloidogyne incognita* nematode	Tomato/root knot nematode	Amended in soil	*In vivo*	Laquale et al., 2015 [26]
*Rhizopus stolonifera* zygomycota	Peach fruit/soft rot decay	Fumigation	*In vitro* and *in vivo*	Zhou et al., 2018 [27]
*Penicillium expansum* fungus	Pear fruit/blue mold	Fumigation	*In vitro* and *in vivo*	Neri et al., 2006 [28]
*Penicillium* spp. fungus	NA	Agar diffusion	*In vitro*	Scora and Scora, 1998 [29]
*Escherichia coli*/*P. digitatum* bacterium/fungus	Blueberry fruit	Coated onto fruit	*In vitro* and *in vivo*	Sun et al., 2014 [30]
*Monilinia fructicola*, *Botrytis cinereal* fungus	Apricot fruit/brown and gray mold rot	Sprayed onto fruit	*In vivo*	Hassani et al., 2012 [31]

**Table 2 molecules-24-01932-t002:** Effect of carvacrol on pathogenic fungi and bacteria in media.

Carvacrol Concentration (mM)	Mycelial Growth Rate on Media (mm/day) ^z^		*Xanthomonas Perforans* Concentration after Incubation in Nutrient Broth (CFU/mL) ^y^		
	*Fusarium* Spp.	*Rhozoctonia Solani*	30 min	1 h	2 h
0	3.5 ^a^	10.6 ^a^	2.9 × 10^6^	5.2 × 10^6^	4.4 × 10^6^
0.25	2.3 ^b^	0.9 ^b^	2.1 × 10^6^	3.3 × 10^6^	3.9 × 10^6^
0.5	0	0	6.5 × 10^5^	1.2 × 10^6^	3.5 × 10^5^
0.75	0	0	1.6 × 10^2^	15	0
1.0	0	0	0	0	0

^z^ Potato dextrose agar (PDA) plates amended with carvacrol were prepared the same day when they were inoculated with pathogens. Means followed by the same letter in the same column were not significantly different at *P* = 0.05. ^y^ Nutrient broth of 20 mL amended with carvacrol was inoculated with 200 µL of bacterial cells grown on nutrient agar (NA) for 24 h (1 × 10^9^ CFU/mL) and incubated at 28 °C with continuous shaking (rpm = 110). A 10-time serial dilution was conducted to determine bacterial concentrations at each sampling time, in which a 20 µL of solution with three replicates was dropped onto NA media for each dilution. ‘0’ CFU/mL means no bacterial colony was detected from the original solution.

**Table 3 molecules-24-01932-t003:** Foliar application of carvacrol on squash powdery mildew (*Podosphaera xanthii*) in greenhouse.

Treatment	Timing of Application ^z^		Disease Severity (%) ^y^
	2 h before Inoculation	24 h after Inoculation	
Untreated control	NO	NO	64 ^a^
Carvacrol at 0.1 mM	YES	NO	59.0 ^a,b^
	YES	YES	49.0 ^c^
Carvacrol at 0.5 mM	YES	NO	61.5 ^a,b^
	YES	YES	36.5 ^d^
Carvacrol at 1.0 mM	YES	NO	58.0 ^a,b^
	YES	YES	40.5 ^d^
Carvacrol at 5.0 mM	YES	NO	56.0 ^b^
	YES	YES	37.0 ^d^

^z^ First two fully expanded true leaves of each squash plant were treated with carvacrol and inoculated with a spore suspension at 2 × 10^3^ spore/mL. ^y^ Percentage of leaf area covered by colonies of powdery mildew was rated as disease severity for each inoculated leaf. Experiment was conducted twice and results were combined for data analysis after no significant difference between two repeats was detected. Means followed by same letter in the column were not significantly different at *P* = 0.05.

**Table 4 molecules-24-01932-t004:** Foliar application of carvacrol on disease severity of bacterial spot, plant vigor and yield of tomato in the field.

Treatment	Early Blight (%) ^zw^	Bacterial Spot (%) ^zw^	Vigor ^yw^	Yield of Extra and Large Fruit (kg/plant) ^xw^	Yield of Total Fruit (kg/plant) ^xw^
Untreated control	38.1 ^a^	6.9 ^a^	3.2 ^b^	0.67 ^b^	1.65 ^b^
Carvacrol @ 0.5 mM	32.0 ^b^	2.4 ^b^	4.3 ^a^	1.23 ^a^	2.13 ^a^
Carvacrol @ 1.0 mM	30.6 ^b^	3.5 ^b^	4.5 ^a^	1.27 ^a^	2.22 ^a^

^z^ Disease severity of early blight (*Alternaria tomatophila*) and bacterial spot (*X. perforans*) was recorded as the percentage leaf area of the whole plant with symptoms of each disease. ^y^ Vigor of the plant was rated based on a 1-5 scale: 5= vigorously growing; 4 = slight yellowing and a little short; 3 = more yellowing and shorter; 2 and 1 = growing very poor than normal. ^x^ Tomato fruit were harvested from each plant and fruit size was graded as extra-large, large, and medium. ^w^ Means followed by the same letters in each column were not significantly different at *P* = 0.05.

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
