# Peer review of "Potential of a Small Molecule Carvacrol in Management of Vegetable Diseases"

_molecules, 2019, doi:10.3390/molecules24101932_

Round 1

Reviewer 1 Report

The paper is clear, well written and results anddiscussion are well supported by the experimetal design. Comments are reported directly in the revisione enclosed version

Author Response

Thanks for your suggestion for improving the manuscript. Below is the responses regarding changes that have been made.

Changes have been made according to the suggestions on:

                P2, L37-43;          P5, L103-104;      P6, L114-116;

Formation changes have also been made for the tables.

Footnote for the figure on P4 was not changed as the information was critical regarding how the experiments were conducted.        

Reviewer 2 Report

Suggestions:

1) The chemical structure of carvacrol is not needed. Therefore, the figure 1 should be removed.

2) If allowed (limit of words), provide more background of the study.

3) Section 2 should be improved.

4) A conclusion section is expected. It should present the reality and the future expectations as well.

5) Present a key conclusion in the abstract.

6) Provide more discussion behind the data presented in the tables.

Author Response

Thank you for the suggestions in improving the manuscript. Below are changes that have been made according to your suggestions.

Suggestions:

1)     The chemical structure of carvacrol is not needed. Therefore, the figure 1 should be removed.

    Figure 1 has been removed as suggested.

2)     If allowed (limit of words), provide more background of the study.

    Additional information has been provided regarding the studies on carvacrol (P2, L60-62).

3)     Section 2 should be improved.

    Modifications have been made in this section.

4)     A conclusion section is expected. It should present the reality and the future expectations as well.

    A paragraph of conclusion on studies with carvacrol has been specified according to this suggestion. (P8, L172-177).

5)     Present a key conclusion in the abstract.

    A sentence of conclusion has been added in the abstract. (P1, L21-24)

6)     Provide more discussion behind the data presented in the tables.

    More discussions have been provided regarding the findings with carvacrol (P4, L69-74; P5, L94-97; P6, L125-127).